# Factors Limiting the Growth of Eucalyptus and the Characteristics of Growth and Water Use under Water and Fertilizer Management in the Dry Season of Leizhou Peninsula, China

**Zhichao Wang [1,2], Apeng Du [1,2,*], Yuxing Xu [1,2], Wankuan Zhu [1,2] and Jing Zhang [1]**

1   China Eucalypt Research Centre (CERC), Chinese Academy of Forestry (CAF), Zhanjiang 524022, China
2   Guangdong Zhanjiang Eucalyptus Plantation Ecosystem Research Station, Zhanjiang 524022, China
*   Correspondence: cercdap@caf.ac.cn; Tel.: +86-759-338-2163; Fax: +86-759-338-0921

**Abstract:** The growth rate of eucalyptus in the dry season was significantly lower than that in the wet season. However, the limiting factors of eucalyptus growth in the dry season are not clear. In this paper, through the continuous monitoring of the diameter growth and environmental factors of 5.5-year-old *Eucalyptus urophylla* S. T. Blake × *E. grandis* W. Hill ex Maiden in the dry season, the diameter growth characteristics of eucalyptus during the dry season were studied and the limiting factors of eucalyptus growth in the dry season were determined. The water and fertilizer management activities in the dry season were evaluated to verify the growth and water use characteristics of *Eucalyptus urophylla* × *E. grandis* in the dry season under the conditions of mitigation limiting factors and provide the basis for further increasing the growth rate of eucalyptus. The results show that the diameter fluctuation of *Eucalyptus urophylla* × *E. grandis* is cyclical and the diameter cumulative growth during the dry season monitoring is consistent with the Gompertz model. Atmospheric temperature and soil water content are the main factors limiting the growth of *Eucalyptus urophylla* × *E. grandis* in the dry season. Irrigation and fertilization in the dry season can significantly increase the growth of diameter at breast height (DBH) and biomass growth and significantly improve the water use efficiency in the dry season.

**Keywords:** diameter growth; *E. urophylla* × *E. grandis*; dry season; limiting factor; water and fertilizer management; water use efficiency

## 1. Introduction

China is the world's largest timber importer [1,2] and the situation of timber supply security is severe. The demand for wood in China is expected to reach 800 million m$^3$ by 2020. In recent years, the shortage of international timber resources [3,4], coupled with the complete cessation of commercial harvesting of natural forests in China, has made China's timber dependence on imports to nearly 60% [5]. Plantations play an important role in forest resources and an increasingly important role in wood production, environmental improvement and climate change mitigation [6]. As one of the three fast-growing trees genera recognized in the world, the current plantation area of eucalyptus in China is about 4.5 million hectares [7]. Although it only accounts for 6.5% of China's planted forest area, the annual output of wood exceeds 30 million m$^3$, accounting for 26.9% of the national timber production [5]. Therefore, eucalyptus has become an important strategic tree genus in southern China.

However, the average annual growth of eucalyptus commercial lumber production forests in China is only 10–30 m$^3$/ha, which is much lower than other countries such as Brazil (40 m$^3$/ha) and South Africa (35 m$^3$/ha) [8–10]. Affected by climatic conditions, the planting area of eucalyptus in

China has a distinct dry and wet season [11–13]. Taking the Leizhou Peninsula as an example, the annual precipitation is concentrated from May to October, accounting for 77%–85% of the annual precipitation, with typical dry and wet seasons. Many studies have shown that the growth rate and transpiration water consumption of eucalyptus in the dry and wet seasons in China are quite different. The growth of a 2-year-old *Eucalyptus urophylla* × *E. grandis* plantation in the wet season is more than 4 times that of the dry season [14]. The average volume increment per tree of 1-5-year-old eucalyptus in Hainan Province during the dry season (0.0023 m$^3$) was only 37.4% of that in the wet season (0.0061 m$^3$) [15]. The average daily water consumption in the wet season of the 10-year-old *Eucalyptus urophylla* plantation in Leizhou Peninsula is 1.91 mm/d, which is 1.5 times the average daily water consumption in the dry season [16]. Also, in the Leizhou Peninsula, the average monthly flux of sap flow in a 2-3-year-old *Eucalyptus urophylla* × *E. grandis* tree in the wet season was 122.4 kg/month, which was 1.53 times higher than that in the dry season [17]. The survival of trees was also largely controlled by water supply under certain temperature conditions. Water stress was the most common and important among the various factors that limit the yield of trees to achieve their genetic potential [18,19]. Eucalyptus will alleviate drought stress by improving water use efficiency if it is under water stress [20]. Some related studies have proved that the water use efficiency of eucalyptus in the dry season is significantly higher than that of the wet season in the planting areas in China [14,21]. This leads to a series of scientific questions. What are the limiting factors for the growth of eucalyptus in the dry season of the planting areas in China? Is it mainly limited by water conditions? Can we increase the growth rate of eucalyptus through dry season irrigation? In addition, fertilization is one of the indispensable forest management measures in view of the short-rotation of eucalyptus forests [22], if irrigation is carried out in the dry season and fertilization is applied at the same time, can the water use efficiency of eucalyptus forest be further improved, thereby increasing the yield per unit area of wood?

The dendrometer (DC3, Ecomatik, Munich, Germany,) is currently used to monitor growth rates on short time scales and then the response of stem diameter growth to environmental factors can be analyzed [23]. We conducted an experiment in a 5.5-year-old *Eucalyptus urophylla* × *E. grandis* plantations in the Leizhou Peninsula characterized by a pronounced dry season. We tested four treatments during the dry season; a non-treated control, irrigation alone, fertilization alone and irrigation and fertilization combined during the January to April dry season. Two methods were used to continuously measure the diameter growth of eucalyptus and the meteorological factors were simultaneously measured to analyze the response of eucalyptus growth to environmental factors in the dry season of Leizhou Peninsula. The water use efficiency of eucalyptus under different water and fertilizer managements in the dry season was compared and the theoretical basis and data support were provided to improve the wood production of eucalyptus in the dry season.

## 2. Materials and Methods

### 2.1. Study Site and Materials

The study area is located in the Leizhou Peninsula area of Guangdong province in China. The experimental sites are located in Guangdong Zhanjiang Eucalyptus Plantation Ecosystem Research Station (21°30′ N, 111°38′ E), with an elevation of 90 m (Figure 1). The terrain is flat and belongs to the maritime monsoon climate. The annual average rainfall is about 1500 mm, mostly concentrated in May to October, accounting for 77%–85% of the annual rainfall (Figure 2).

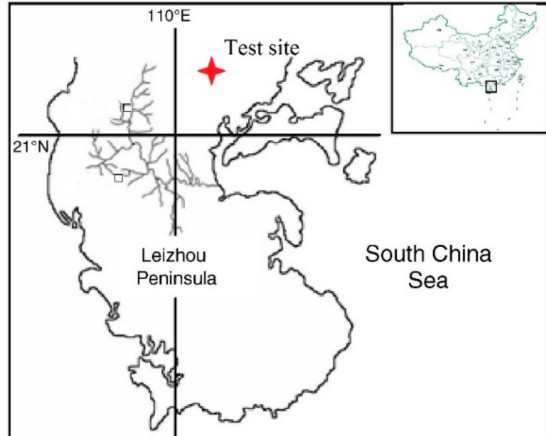

**Figure 1.** Leizhou Peninsula, western Guangdong province, showing the location of the monitored plantations at the red star symbol.

The two-ha experimental forest of *Eucalyptus urophylla* × *E. grandis* clone 32–29, was planted in July 2012 at 1666 plants ha. The afforestation method is burrowing afforestation (length 50 cm × width 50 cm × depth 40 cm). The existing density is 960 plants·ha. In the initial stage of monitoring, that is, at the end of the last rainy season, the average diameter at breast height (DBH) of the stands was 14.41 cm, the average tree height was 15.8 m and the average LAI (leaf area index) was 1.35.The soil types in the experimental sites are mainly Basalt-Latosols, the thickness of the soil layer is more than two meters, the average organic matter content in the 0–80 cm soil layer is more than 1.6%, the pH is 5.0 and the soil fertility is at a medium level.

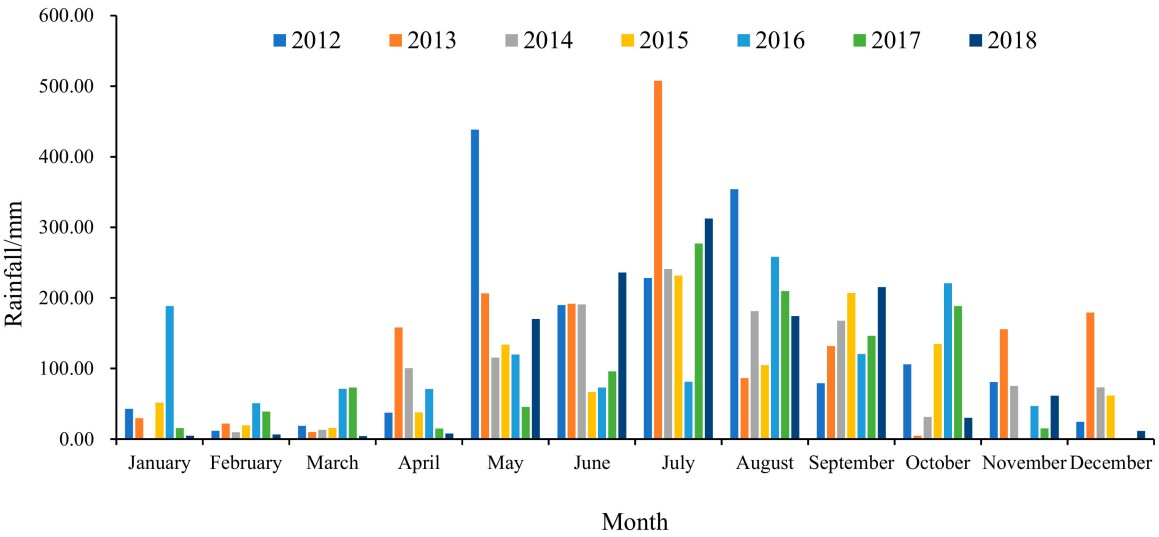

**Figure 2.** Precipitation distribution in each month from 2012 to 2018.

## 2.2. Research Method

(1) Experimental design

We randomly divided two hectares of the 5.5-year-old *Eucalyptus urophylla* × *E. grandis* stand into four small blocks, then from January 16 to April 18, four treatments were carried out: irrigation and fertilization (monthly water supply is about 500 L per tree(the same below), fertilizer amount is 500 g per tree (NPK15-5-9,total one time, under Same), fertilization alone, irrigation alone and non-treated control (CK).In each treated forest land, three monitoring plots of 20m × 20m were established.

According to the survey results of each plot, nine trees with similar average breast diameters were selected as sample trees (Table 1).

**Table 1.** Basic conditions of the nine sample trees; the three sample trees of Non-treated control (CK) were installed with the dendrometer; all nine sample trees used for sap flow.

| Treatments | Sample Tree Number | DBH/cm | Tree Height/m | Crown Width (EW/m × SN/m) |
|---|---|---|---|---|
| Non-treated control (CK) | 1 | 14.72 | 16.2 | 3.8 × 4.0 |
| | 2 | 13.21 | 15.2 | 2.5 × 3.0 |
| | 3 | 14.94 | 16.4 | 3.2 × 3.6 |
| Irrigation alone | 1 | 13.78 | 15.3 | 2.6 × 3.2 |
| | 2 | 15.94 | 16.2 | 3.5 × 3.6 |
| | 3 | 14.89 | 16.4 | 3.0 × 3.2 |
| Fertilization alone | 1 | 14.76 | 15.7 | 3.3 × 3.5 |
| | 2 | 13.43 | 16 | 2.8 × 3.1 |
| | 3 | 14.91 | 16.4 | 3.3 × 3.6 |
| Irrigation and fertilization | 1 | 13.52 | 16.2 | 2.1 × 3.7 |
| | 2 | 13.13 | 15.5 | 2.6 × 3.8 |
| | 3 | 16.24 | 17.2 | 3.4 × 4.1 |

(2) Eucalyptus tree stem diameter growth

Because the eucalyptus trees were very tall, it was not easy to monitor the tree height continuously but continuous monitoring of DBH could be achieved. The diameter growth can reflect the growth of the forest to a certain extent [24,25]. The three sample trees of the control plot were installed with the dendrometer (Figure 3A) for continuous monitoring of the variation of stem diameter growth according to the results of the control registration. The data of stem diameter growth was collected by using a data collector (HOBO, Onset Data1-800-Loggers, onset, Cape Cod, America) with a monitoring frequency of 30 min.

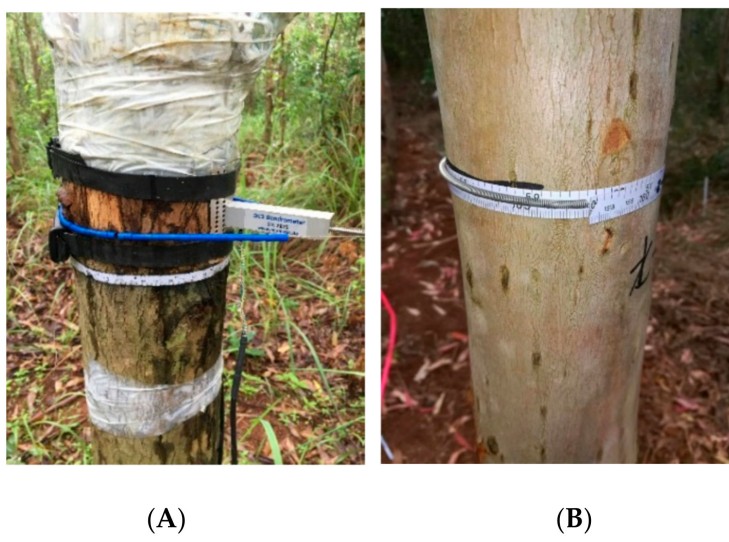

(**A**)                    (**B**)

**Figure 3.** Two kinds of tree diameter measuring device diagram: (**A**) the dendrometer (DC$_3$); (**B**) the self-made growth rings.

In addition, the self-made growth rings were installed at 1.3 m height of each tree in each of the monitoring plots in each of the four treated areas (Figure 3B). Through regular surveys twice per month, The DBH growth of *Eucalyptus urophylla* × *E. grandis* under different treatments was monitored from January to April in 2018.

(3) Sap flow and meteorological factors

In each of the treated plots, the sap flow velocity of *Eucalyptus urophylla* × *E. grandis* trees were continuously monitored by the Granier thermal diffusion method on selected sample trees (Table 1). Specific methods are as follows—the SF-G type (two needles, length of probe 33 mm) thermal diffusion probes of Ecomatic Company of Germany were installed at 1.3 m of each selected sample tree and the data were collected by CR3000 data collector of Campbell Company of America, with the collection interval of 30 minutes. In order to avoid the difference of sap flow in different directions and prevent the effects of sunlight, the probes were installed on the shade of the tree stem and covered with radiation-proof aluminum foil.

Meanwhile, one portable automatic weather station (WATCH DOG-2900ET) was set up in the experimental forest and an automatic weather station (Campbell-CR3000) was set up in the open area near the test forest. The meteorological factors in and outside the experimental forest, mainly including Atmospheric temperature (T), Air humidity (RH), Photosynthetically available radiation (PAR), Soil temperature (T), Wind speed, Wind direction, Precipitation (P), Water surface evaporation and other indicators were measured (Table 2). The data collection interval was set to ten minutes. In addition, in order to consider the synergistic effect of atmospheric temperature and humidity, the VPD (Air saturation vapor pressure deficit, Kpa) was calculated by using atmospheric temperature and relative humidity. The formula was as follows:

$$VPD \ = \ (1-RH) \times 0.6108e\left(\frac{17.27T}{T+273.3}\right) \tag{1}$$

where *RH* is the relative air humidity (%). The *T* is the air temperature (°C).

(4) Sapwood area and individual biomass

Based on the survey of the experimental forest, nine analytic trees were selected according to the three diameters classes. A relationship equation between sapwood area and DBH is established by measuring the breast diameter (DBH), the thickness of the sapwood and the thickness of the bark of the selected trees (once in east-west direction and once in North-South direction) (Figure 4A). Then the sapwood area of sample trees used for sap flow and the whole stand is calculated by using the established relation equation. The equation is obtained by regression:

$$As \ = \ 0.8619DBH^2 - 12.466DBH + 74.547 \qquad R^2 = 0.9506 \tag{2}$$

where *AS* is the sapwood area ($cm^2$). The *DBH* is the diameter at breast height (cm).

In addition, the regression equations of DBH and biomass were obtained by the method of tree stem analysis. (Branch, foliage, stem, bark and stemwood of the analytic trees were all weighed for fresh weight and samples of each biomass component were taken and weighed respectively. Finally, samples of each component were brought back to the laboratory to be dried and the sample biomass was obtained. Then, biomass of each component was calculated according to the mass ratio of the sample and the whole tree. Then the increment of DBH was used to calculate the increment of individual biomass (Figure 4B).

**Table 2.** Meteorological overview during the study period (2018.1.16–2018.4.18).

| Item | Atmospheric Temperature °C | Air Relative Humidity % | Wind Speed m·s$^{-1}$ | Daily Average PAR μmol·(s$^{-1}$·m$^{-2}$) | Rainfall mm·d$^{-1}$ | Water Surface Evaporation mm | VPD Kpa |
|---|---|---|---|---|---|---|---|
| Mean | 19.50 | 79.58 | 5.75 | 112.43 | 0.25 | 0.34 | 0.42 |
| Minimum | 5.64 | 40.73 | 1.21 | 18.20 | 0.00 | 0.09 | 0.04 |
| Maximum | 27.37 | 98.10 | 12.23 | 217.50 | 2.60 | 5.08 | 0.77 |

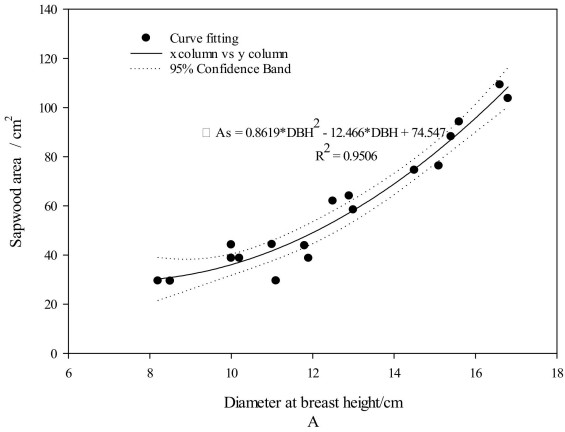

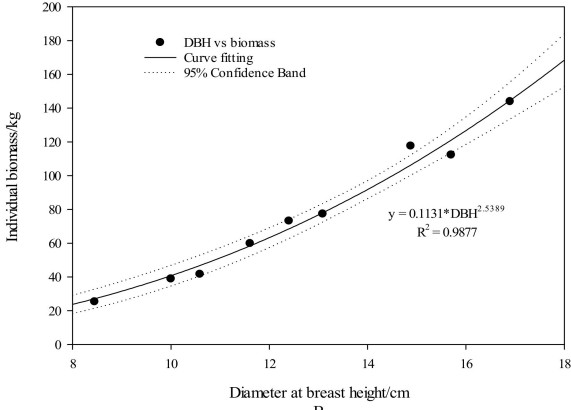

**Figure 4.** Fit curve of diameter at breast height (DBH) and sapwood area (**A**), DBH and biomass (**B**).

### 2.3. Statistical Analysis

The sap flow velocity is calculated according to the general Granier's sap flow formula [26].

$$Js = 0.0119K^{1.231} \times 60; K = (\Delta Tmax - \Delta T)/\Delta T \tag{3}$$

where Js refers to the sap flow velocity (cm·min$^{-1}$), $\Delta Tmax$ is the temperature (°C) of the heated probe, obtained when Js approaches zero. $\Delta T$ is the temperature difference (°C) between heated and not-heated needles.

Then, the sap flow flux of each *Eucalyptus urophylla×E. Grandis* tree is calculated by the following equation.

$$Fi = Js \times As \times 10^{-3} \times t \tag{4}$$

where Fi refer to the sap flow flux, AS is the sapwood area (cm$^2$), t is the Corresponding time.

The daily diameter growth is calculated by the maximum method [27]: that is the difference between the maximum values of stems in the adjacent two days is calculated on sunny days and the corresponding correction formula is adopted for the rainfall weather [24,28,29].

$$Gd = \begin{cases} D_{\max(i+1)} - D_{\max(i)} & (0mm < P_d < 10mm) \\ \frac{1}{n}\left(D_{\max\alpha} - D_{\max\beta}\right) & (10mm \leq P_d) \end{cases} \tag{5}$$

where the daily diameter growth (μm), $D_{\max (i)}$ and $D_{\max (i + 1)}$ are respectively the stem maximums of the *i* and *i* + 1 days (μm), $D_{\max \alpha}$ and $D_{\max \beta}$ are respectively the maximum stem diameter before and

after rainfall (μm), n is the number of rainy days. The cumulative diameter growth of stem refers to the sum of the cumulative daily diameter growth.

A primary statistical analysis and the graphic drawing was carried out using the Sigma Plot 13.0 software and Excel (2016). Regression analysis of biomass and DBH, sapwood area and DBH, diameter cumulative growth and time were performed by spss19.0. One-way ANOVA was performed using spss19.0 software for the DBH growth, cumulative growth, biomass increase, sap flow velocity and water use efficiency of different treatments. Pearson correlation analysis and partial correlation analysis were used to evaluate the relationship between the cumulative diameter growth and environmental factors. Partial correlation analysis can exclude the mutual influence among the influencing factors.

## 3. Results

### 3.1. Diameter Diurnal Variation and Cumulative Change Dynamics During Dry Season

The data of 5 days from 4 April to 8 April 2018 was selected to show day and night changes of the diameter growth for *Eucalyptus urophylla × E. grandis* stems under typical weather conditions during the dry season (Figure 5) and rainy weather (rainfall 5.8 mm) on 7 April. On sunny days, the diameter diurnal fluctuation of *Eucalyptus urophylla × E. grandis* is in a significant periodic pattern and from this pattern three distinct phases (Shrinkage stage, Recovery stage and Increment stage) were defined within a single diurnal cycle. The maximum value of stem diameter growth appeared at 09:00 and the minimum occurred at 18:00, this periodic variation of stem diameter growth on the daily scale is a reversible biological process, possibly due to the action of water in the stem. In rainy days, the change of diameter growth is not obvious, this may be due to the lack of solar radiation, which results in a weaker transpiration rate throughout the day. In this case, the transpiration rate is always lower than the root water absorption rate and the diameter expansion continues until the diameter stem reaches water saturation [30].

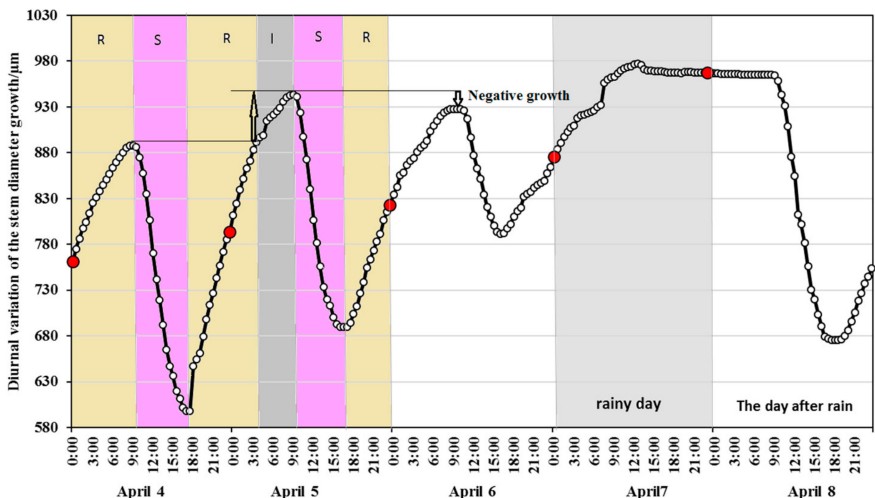

**Figure 5.** A typical diurnal cycle during April 2018. The time of commencement and cessation of the phases of shrinkage (S), recovery (R) and Increment (I) are determined.

We found that the diameter variations on trees in the control plot at different times are quite different (Figure 6).The stem diameter growth during the whole monitoring period (from January to April 2018) can be divided into three stages, namely "Relatively stability period," "Rapid growth period" and "Dehydrate contraction period." From January 25 to February 7, which was a Relatively stable period, the diameter variation remained relatively stable and the maximum diameter variation of *Eucalyptus urophylla × E. grandis* trees was only 59.54 μm. From February 8 to March 13, which was a rapid growth period (33 days), the average daily stem diameter growth increased to 42.89 $\mu m \cdot d^{-1}$ and the total growth was 1458. 38 um. The maximum daily growth value was 69.4 $um \cdot d^{-1}$, which

appeared on February 13. From March 13 to the beginning of the wet season, the stem diameter growth rate of *Eucalyptus urophylla* × *E. grandis* entered a slow stage once again, with a diameter variation of −195.12 μm. At this stage, the stem diameter growth of *Eucalyptus urophylla* × *E. grandis* became negative.

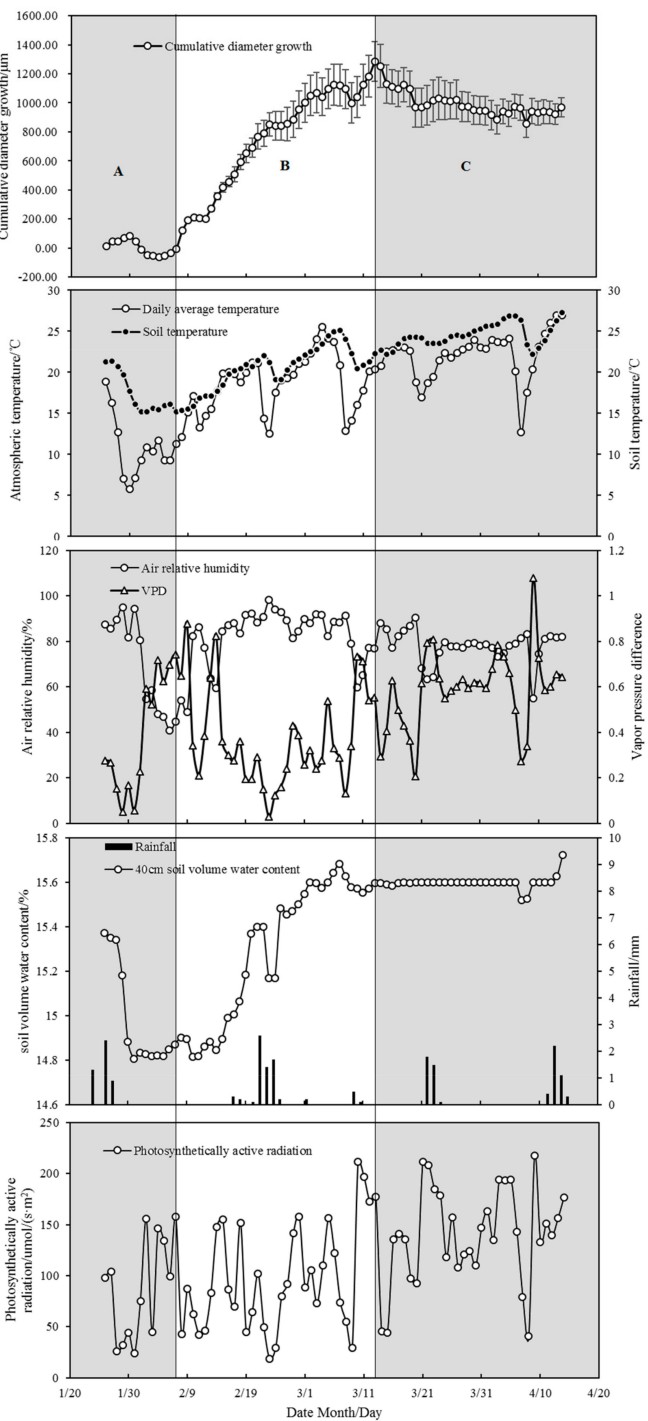

**Figure 6.** Diameter cumulative growth characteristics and dynamic changes of related environmental factors of *E. urophylla* × *E. grandis* on trees in the control plot during monitoring in dry season. The diameter cumulative growth throughout the dry season was divided into three phases: **A**: Relatively stable period; **B**: Rapid growth period; **C**: Dehydrate contraction period.

Using the Gompertz model to curve the stem diameter growth rate of *Eucalyptus urophylla* × *E. grandis* in the dry season, the following fitting model can be obtained [31–33].

$$y = 1004.43exp\left(-e^{(8.37 - 0.1885t)}\right) \quad R^2 = 0.962 \quad P < 0.01 \tag{6}$$

where *y* is the cumulative stem diameter growth (μm), t is the corresponding time.

By deriving this fitted model, the stem diameter growth rate of *Eucalyptus urophylla* × *E. grandis* at each time point during the monitoring can be obtained (Figure 7). According to Figure 7, the stem diameter growth rate of *Eucalyptus urophylla* × *E. grandis* trees in dry season showed a single peak curve during the monitoring period and the stem diameter growth rate in the "Relatively stability period "and the "Dehydrate contraction period" was very close to zero.

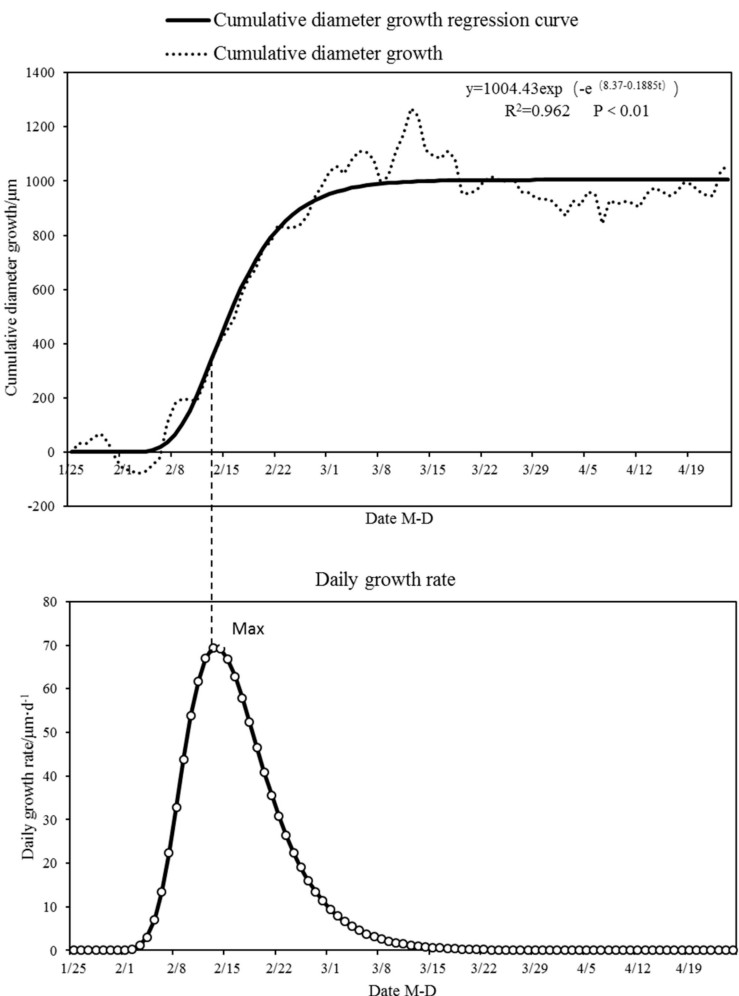

**Figure 7.** Diameter cumulative growth curve fitting and daily growth rate of *E. urophylla* × *E. grandis* on the control plot in dry season.

*3.2. Analysis of Limiting Factors for Diameter Growth in Dry Season*

In addition to the biological characteristics of the tree species, the growth rate of trees is also closely related to the site environment and meteorological conditions [34,35]. From Figure 6, the trend of atmospheric temperature, soil temperature, air relative humidity, saturation vapor pressure deficit and soil moisture with time were consistent with the trend of stem diameter growth of *Eucalyptus urophylla* × *E. grandis* trees during the study period. Through Pearson correlation analysis we found that the cumulative stem diameter growth of *Eucalyptus urophylla* × *E. grandis* during the study

period was significantly positively correlated with atmospheric temperature, air relative humidity, photosynthetically available radiation, soil moisture and soil temperature ($p < 0.01$), with correlation coefficients of 0.744, 0.359, 0.347, 0.892 and 0.81,respectively (Table 3). However, in order to exclude the interaction between meteorological factors, partial correlation analysis shows that the cumulative stem diameter growth of *Eucalyptus urophylla* × *E. grandis* was significantly positively correlated with soil moisture ($p < 0.01$) with a correlation coefficient 0.623 and with atmospheric temperature ($p < 0.05$) with a correlation coefficient of 0.282. There was no significant correlation with air relative humidity, photosynthetically active radiation and soil temperature (Table 3).

**Table 3.** Analysis of influence factors of diameter growth of *E. urophylla* × *E. grandis* in dry season.

| Environmental Factors | Pearson Correlation Analysis | | Partial Correlation Analysis | |
|---|---|---|---|---|
| | Pearson Correlation Coefficient | P | Partial Correlation Coefficient | P |
| Atmospheric temperature | 0.744 ** | 0.000 | 0.282 * | 0.014 |
| Relative humidity of air | 0.359 ** | 0.001 | 0.103 | 0.380 |
| Wind speed | −0.022 | 0.847 | | |
| Photosynthetically active radiation | 0.347 ** | 0.002 | 0.019 | 0.870 |
| Rainfall | −0.041 | 0.718 | | |
| Soil volumetric water content | 0.892 ** | 0.000 | 0.623 ** | 0.000 |
| Soil temperature | 0.81 ** | 0.000 | −0.122 | 0.296 |
| Vapor pressure difference | 0.184 | 0.333 | | |

The interactions among the main factors include ** $p < 0.01$, * $p < 0.05$ and ns as not significant.

It can be seen that in the seasonal dry climate of Leizhou Peninsula, the most important factor limiting the stem diameter growth of *Eucalyptus urophylla* × *E. grandis* trees was soil moisture, followed by atmospheric temperature. Finally, by regression analysis of the two main factors (atmospheric temperature and soil moisture) affecting the cumulative stem diameter growth of *Eucalyptus urophylla* × *E. grandis*, the fitting equation of the cumulative stem diameter growth of *Eucalyptus urophylla* × *E. grandis* trees can be obtained.

$$y = -14806.751 + 991.528SW + 15.818T \quad R^2 = 0.813 \quad p < 0.01 \tag{7}$$

where $y$ is the cumulative stem diameter growth (μm), $SW$ is the soil moisture (%), $T$ is the atmospheric temperature (°C).

### 3.3. Characteristics of Growth and Water Use Under Water and Fertilizer Management in Dry Season

The average DBH growth under the irrigation and fertilization treatment (0.291 cm) was significantly higher than that under other three treatments during the monitoring period ($p < 0.05$) and there were no significant differences between the remaining 3 treatments (Table 4). There were no differences between the control and the three other treatments (irrigation alone, fertilization alone, irrigation and fertilization) for the Relatively stable period and Dehydrate contraction period. In the Rapid growth period, the average DBH growth under the irrigation and fertilization treatment (0.168 cm) was significantly higher ($p < 0.05$) than the irrigation alone treatment (0.134 cm) and CK (0.111 cm) but not significantly different with the fertilization treatment (0.151 cm).

**Table 4.** The DBH growth under different treatments of *E. urophylla* × *E. grandis* in the dry season monitoring period. The numbers show the treatment means ± SE of the mean. Means with different lowercase letters are significantly different ($p < 0.05$).

| Growth Attribute | Treatments | Different Growth Stages | | |
|---|---|---|---|---|
| | | 1/16–2/03 (Relatively Stable Period) | 2/03–3/14 (Rapid Growth Period) | 3/14–4/18 (Dehydrate Contraction Period) |
| Breast diameter increase | Irrigation and fertilization | 0.093 ± 0.006 a | 0.168 ± 0.011 a | 0.03 ± 0.024 a |
| | Irrigation alone | 0.075 ± 0.013 ab | 0.134 ± 0.025 b | −0.029 ± 0.027 a |
| | Fertilization alone | 0.054 ± 0.003 b | 0.151 ± 0.008 ab | 0.006 ± 0.012 a |
| | Non-treated control (CK) | 0.073 ± 0.012 ab | 0.111 ± 0.014 b | −0.02 ± 0.014 a |
| Breast diameter cumulative growth | Irrigation and fertilization | 0.093 ± 0.006 a | 0.261 ± 0.017 a | 0.291 ± 0.028 a |
| | Irrigation alone | 0.075 ± 0.013 ab | 0.209 ± 0.035 ab | 0.206 ± 0.025 b |
| | Fertilization alone | 0.054 ± 0.003 b | 0.205 ± 0.01 ab | 0.211 ± 0.018 b |
| | Non-treated control (CK) | 0.073 ± 0.012 ab | 0.184 ± 0.02 b | 0.164 ± 0.03 b |

There were a differences in the daily average sap flow velocity of the *Eucalyptus urophylla* × *E. grandis* individual tree among four treatment plots in three stages during the study period (Figure 8). In the Rapid growth period, the daily average sap flow velocity in the irrigation and fertilization plot (0.0584 cm·min$^{-1}$) was significantly higher than that of the other three plots (irrigation alone, 0.0496 cm·min$^{-1}$; fertilization alone, 0.0491 cm·min$^{-1}$; CK, 0.0471cm·min$^{-1}$). There was no significant difference in the daily average sap flow velocity among the four plots in the other two periods.

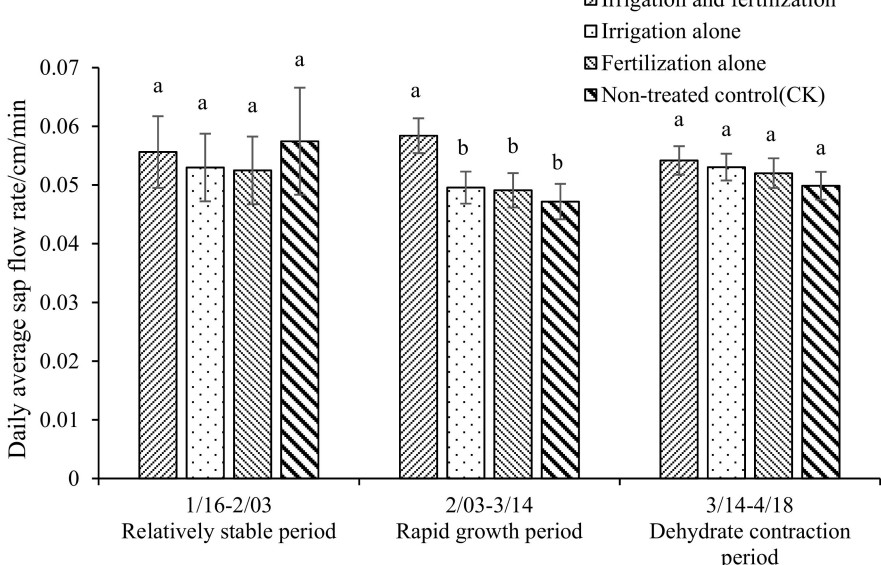

**Figure 8.** The sap flow velocity under different treatments of *E. urophylla* × *E. grandis* in the dry season monitoring period. The bars show the treatment means ± SE of the mean. Means with different lowercase letters are significantly different ($p < 0.05$).

Water use efficiency of *Eucalyptus urophylla* × *E. grandis* in the irrigation and fertilization plot (9.90 g·L$^{-1}$) was significantly higher than that of other three plots (irrigation alone, 7.09 g·L$^{-1}$; fertilization alone, 7.09 g·L$^{-1}$, CK, 6.21 g·L$^{-1}$) during the study period (from January 16 to April 18 in 2018). In the period from January 16 to February 3 and the Rapid growth period (from February 3 to March 14), the water use efficiency of the other three treatment plots did not significantly differ with the CK. During the Dehydrate contraction period (from March 14 to April 18), the water use efficiency in the irrigation and fertilization plot (4. 13 g·L$^{-1}$) was significantly higher than that of other three plots, a similar result as observed for the whole study period (from January 16 to April 18 in 2018).

Due to the very small or even negative diameter growth in Dehydrate contraction period (from March 14 to April 18), the water use efficiency of *Eucalyptus urophylla* × *E. grandis* in this period was much lower than that in other two periods (Table 5).

**Table 5.** Water use efficiency under different treatments at different growth stages. The numbers show the treatment means ± SE of the mean. Means with different lowercase letters are significantly different ($p < 0.05$).

| Treatments | Different Growth Stages | | | |
|---|---|---|---|---|
| | 1/16–2/03 (Relatively Stable Period) | 2/03–3/14 (Rapid Growth Period) | 3/14–4/18 (Dehydrate Contraction Period) | Total |
| Irrigation and fertilization | 15.10 ± 0.991 a | 12.77 ± 0.694 a | 4.13 ± 0.99 a | 9.9 ± 0.964 a |
| Irrigation alone | 13.99 ± 1.758 a | 12.97 ± 1.641 a | 0.22 ± 0.146 b | 7.09 ± 0.939 b |
| Fertilization alone | 8.64 ± 0.505 b | 12.86 ± 0.613 a | 1.51 ± 0.355 b | 7.09 ± 0.578 b |
| Non-treated control (CK) | 11.75 ± 1.845 ab | 10.79 ± 1.249 a | 0.67 ± 0.355 b | 6.21 ± 1.076 b |

## 4. Discussion

The diameter growth of trees includes the division and elongation of stem cambium cells and the diurnal variation (periodic contraction and expansion) [36]. Similar to *Eucalyptus nitens* (Deane & Maiden) Maiden [37] and other tree species [38–40], the diurnal variation of *Eucalyptus urophylla* × *E. grandis* trees at the daily scale in this study is due to the diameter contraction and expansion caused by the change of stem moisture. The positive and negative relationship between the water absorption rate of the roots and the transpiration rate directly determines the value and duration of stem diameter contraction and expansion [28,41–43]. At the daily scale, the water absorption rate variation of roots was small, while the transpiration rate was a regular periodic fluctuation with high day and low night [17] and the transpiration rate in rainy days was significantly smaller than in sunny days [44]. Therefore, the diurnal variation of *Eucalyptus urophylla* × *E. grandis* was a regular expansion-contraction-expansion fluctuation, while the rainy day pattern was characterized by continuous expansion and then remains unchanged after water saturation. This change is reversible [36,45]. The stem diameter growth by division and elongation of the cambium cells is an irreversible biological process [46–48]. However, the actual growth of cambium in a short time tends to be smaller than the expansion and contraction caused by reversible water changes in the stem [49,50]. And there is no effective method to distinguish them accurately [51]. Therefore, it is more difficult to study the net diameter growth of stems in a short-time than that in a long-time scale.

In forest ecosystems, climate change and various ecological factors directly or indirectly affect the growth process of trees, resulting in corresponding diameter growth changes [52]. At the seasonal level, the dominant environmental factors of stem diameter growth at different stages are different. Similar conclusions have been reported in China and abroad. Biondi et al. observed the radial growth of *Pinus hartwegii* Lindl. at the tropical alpine forest line in Mexico and found that the radial growth of *Pinus hartwegii* was mainly related to soil temperature and solar radiation in the early growing season (April-May). In the middle of the growing season (June-July), the radial growth of trees is mainly affected by soil moisture and the main influencing factors in the late stage are more diversified [53]. In the study of seasonal variation of radial growth of *Larix principis-rupprechtii* Mayr. [31], it was also found that the dominant environmental factors affecting the radial growth of the tree were different in different periods. The daily diameter growth rate of *Eucalyptus urophylla* × *E. grandis* during the study period was quite different, showed a single peak curve as a whole. By analyzing the correlation between the cumulative stem diameter growth and environmental factors during the study period, it can be determined that soil moisture and atmospheric temperature are the main environmental factors limiting the growth of eucalyptus.

Water directly affects the cell division and xylem development of tree cambium [54]. Drought stress will cause the end of tree cambium activity to adapt to the environment [55]. The soil moisture plays an important role in tree transpiration and soil water deficit limits the rate of tree transpiration [56]. During the study period, the diameter growth of eucalyptus in four plots showed a slow-fast-slow intermittent growth pattern, which was consistent with the results of the radial growth characteristics of *E. grandis* ×*E. urophylla* and *E. grandis* ×*E. camaldulensis* Dehnh. Var. *camaldulensis* [57]. Although the growth and water consumption of eucalyptus sample trees in four treatment plots at different stages were different, the diameter growth rate and daily average sap flow velocity of the sample trees in irrigation alone and fertilization alone plots were not significantly different from that of the CK plot during the whole study period, while that of the irrigation and fertilization treatment was significantly higher than that of the CK. This may be because the rapid growth and water consumption of eucalyptus resulted in a soil water deficit only in the period from February 3 to March 14. This is evidenced by the fact that there is no significant difference in the average daily flow rate of the four treated sample trees during the remaining time from February 3 to March 14.

Fertilization in arid environments can exacerbate soil water deficits and thereby limit tree growth [58]. This is demonstrated by the diameter growth of smaller stems treated with fertilization at the initial stage of this study (January 16–February 3). Some studies have shown that there will be a net increase in carbon even when the DBH of trees stops growing at a certain time [59]. In our study (from March 14 to April 18), the diameter cumulative growth of eucalyptus in four treatment plots was very small and even negative in the irrigation alone and the non-treated control plots, which indicated that the stem growth at this stage was in the state of "net carbon increase" regulated by eucalyptus itself or the "latent" state necessary for the next rapid growth. Irrigation alone or fertilization alone could not change this state but the combined effect of water and fertilizer might promote the early recovery of eucalyptus from this state, as evidenced by the fact that the stem diameter growth of the irrigation and fertilization treatment was significantly higher than that of the other three treatments from March 14 to April 18. In this study, an important topic is to verify whether the irrigation can promote the diameter growth of eucalyptus in the dry season. The relevant research results have proved that irrigation alone could not effectively increase the growth of eucalyptus in the dry season. So, the coupling of water and fertilizer is very important. However, in the management of eucalyptus plantations in China, irrigation in the dry season was almost impossible because most of the woodlands were mountainous areas without irrigation conditions. We know that fertilization in the dry season will affect the dissolution and absorption of fertilizers due to soil water deficit. Would fertilization before the end of the wet season or application of liquid fertilizer in the dry season effectively promote the growth of eucalyptus in the dry season? This needs to be verified in the next study.

## 5. Conclusions

The findings of the present study suggest that: the stem diameter growth of *Eucalyptus urophylla* × *E. grandis* is cyclical. Diameter fluctuations on sunny days can be divided into three stages: "Recovery stage," "Shrinkage stage" and "Increment stage." The stems of the rainy days continue to absorb water and remain unchanged after saturation. The diameter accumulation growth during the whole dry season is a volatility rise process, in line with the Gompertz model, which can be divided into 3 growth stages: Relatively stability period, Rapid growth period and Dehydrate contraction period. The main environmental factor limiting the growth of *Eucalyptus urophylla* × *E. grandis* in the dry season is soil moisture, followed by atmospheric temperature. Under the condition that the atmospheric temperature cannot be changed, Iirrigation and fertilization in the dry season can significantly increase the growth of the DBH and biomass in the dry season and improve the water use efficiency of the eucalyptus dry season. Our research results have an important guiding role in optimizing water and fertilizer management measures in eucalyptus plantations during the dry season in China and further improving the annual productivity of eucalyptus.

**Author Contributions:** A.D. and Z.W. conceived and designed the experiments. Z.W. conducted the experiments, performed the analyses and collected the data. Y.X., W.Z. and J.Z. provided the facilities and advised on the preparation of materials. Z.W. wrote the manuscript. Z.W. and A.D. did the statistics evaluations. Z.W. and A.D. read and edited the manuscript. All authors approved the final manuscript.

**Funding:** This research was funded by The National Key Research and Development Program of China(2016YFD0600505), Guangxi Major Science and Technology Project (AA17204087-9), the National Natural Science Foundation of China (31300383).

**Acknowledgments:** We appreciate South China Experiment Nursery for support during the selection of suitable plots of this study.

**Conflicts of Interest:** The authors declare no competing financial interests.

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
