# Peer review of "Factors Limiting the Growth of Eucalyptus and the Characteristics of Growth and Water Use under Water and Fertilizer Management in the Dry Season of Leizhou Peninsula, China"

_agronomy, doi:10.3390/agronomy9100590_

Round 1

Reviewer 1 Report

This manuscript draft is much improved.  Improvements needed to improve grammar and readability.  See marked manuscript with suggested edits and comments.

Reviewer 2 Report

Thank you for addressing my concerns. The manuscript has improved, though I maintain that the study itself is not novel, and the results are tautological.  However, I am sufficiently satisfied with the presentation and and thus cannot reject it.

Author Response

Dear reviewer

     Thank you very much for your professional advice and suggestions on this manuscript, so that this manuscript can be greatly improved. In this revision, based on your feedback, I have made some modifications and improvements to this manuscript (including the introduction). You can see the revised version for details. I hope to get more guidance from you! thank you very much!

Kind regards,

Apeng DU

This manuscript is a resubmission of an earlier submission. The following is a list of the peer review reports and author responses from that submission.

Round 1

Reviewer 1 Report

See attached.

Reviewer 2 Report

The manuscript presents innovative research on factors impacting eucalypt radial growth during the dry season on a site in tropical China.  

The manuscript has the potential to be a solid contribution.  The research topic, methodology, and execution have substantial merit.  

Considerable improvements are needed in the manuscript. Improvement needs include:

1. More detail and clarity in describing methods

2.Eliminate unnecessary statements on methodology in the Introduction and Results section.

3. Improve english expression throughout.

4. Improve table titles and descriptive headings

5. Correct specific problems with several references that are not related to the manuscript topic. 

See the marked manuscript for specific comments, suggestions, and questions.
